# Mitophagy: Molecular Mechanisms, New Concepts on Parkin Activation and the Emerging Role of AMPK/ULK1 Axis

**DOI:** 10.3390/cells11010030

**Published:** 2021-12-23

**Authors:** Roberto Iorio, Giuseppe Celenza, Sabrina Petricca

**Affiliations:** Department of Biotechnological and Applied Clinical Sciences, University of L’Aquila, Via Vetoio, 67100 L’Aquila, Italy; giuseppe.celenza@univaq.it (G.C.); sabrina.petricca@univaq.it (S.P.)

**Keywords:** mitophagy, mitochondria, ubiquitin, PINK1–Parkin pathway, Parkin activation, mitophagy receptors, E3 ligases, AMPK, ULK1

## Abstract

Mitochondria are multifunctional subcellular organelles essential for cellular energy homeostasis and apoptotic cell death. It is, therefore, crucial to maintain mitochondrial fitness. Mitophagy, the selective removal of dysfunctional mitochondria by autophagy, is critical for regulating mitochondrial quality control in many physiological processes, including cell development and differentiation. On the other hand, both impaired and excessive mitophagy are involved in the pathogenesis of different ageing-associated diseases such as neurodegeneration, cancer, myocardial injury, liver disease, sarcopenia and diabetes. The best-characterized mitophagy pathway is the PTEN-induced putative kinase 1 (PINK1)/Parkin-dependent pathway. However, other Parkin-independent pathways are also reported to mediate the tethering of mitochondria to the autophagy apparatuses, directly activating mitophagy (mitophagy receptors and other E3 ligases). In addition, the existence of molecular mechanisms other than PINK1-mediated phosphorylation for Parkin activation was proposed. The adenosine5′-monophosphate (AMP)-activated protein kinase (AMPK) is emerging as a key player in mitochondrial metabolism and mitophagy. Beyond its involvement in mitochondrial fission and autophagosomal engulfment, its interplay with the PINK1–Parkin pathway is also reported. Here, we review the recent advances in elucidating the canonical molecular mechanisms and signaling pathways that regulate mitophagy, focusing on the early role and spatial specificity of the AMPK/ULK1 axis.

## 1. Introduction

Mitochondria are dynamic and multifunctional subcellular compartments forming a sophisticated, interconnected reticulum [1]. They play crucial roles in a range of fundamental processes, including ATP production, heme and steroid hormones biosynthesis, Ca^2+^ metabolism and signaling, iron homeostasis, fatty acid ß-oxidation, the regulation of heterotypic inter-organelle contacts, in particular with the endoplasmic reticulum (ER), and cell death [2,3,4,5,6]. Mitochondrial quality and function must be closely controlled to ensure metabolic substrate availability and to forestall the generation of reactive oxygen species (ROS) and oxidizing agents. Moreover, mitochondria contain ~1000 proteins mainly located in the matrix [7], and the maintenance of proteostasis is ensured by a set of mitochondrial heat shock proteins (HSP), chaperonins, and proteases. In response to particular stressors (including hypoxia, cytokine stimulation, mitochondrial membrane potential alterations, and calcium influx) mitochondria may generate excessive levels of ROS [8,9]. In order to cope with homeostatic insults, cells have evolved multiple interdependent quality-control mechanisms at molecular, organellar and cellular levels that aim to maintain or restore functions [10,11]. The pathways regulating mitochondrial turnover and homeostasis, collectively referred to as mitochondrial quality control (MQC), include DNA repair mechanisms, ROS scavenging, chaperones and proteolytic enzymes, the ubiquitin–proteasome system (UPS), and the mitochondria-specific, unfolded protein response (UPR^mt^). Mitochondrial fusion and fission dynamics, as well as mitochondrial biogenesis and degradation, are also crucial for organelle quality control and maintaining mitochondrial networks [12]. Under stress conditions, mitochondrial fusion and fission events promote content mixing between the compartments that need to be repaired (fusion) and the segregation of damaged organelles (asymmetrical fission) [13,14,15]. Upon alteration in proteostasis, the UPR^mt^ is quickly activated to reduce the buildup of misfolded proteins in the compartment through the transcriptional induction of mitochondrial chaperones and proteases. In particular, UPR^mt^ activation in mammals (including humans) enables the nuclear translocation of activating transcription factor 5 (ATF5), resulting in the expression of its downstream effector genes such as HSP22, HSP60, HSP70, C/EBP homologous protein (CHOP), mtDnaJ, the matrix proteases lon protease homolog (LonP1), and ATP-dependent Clp protease proteolytic subunit (ClpP) [16,17,18]. Several axes of the mammalian UPR^mt^ were reported and their association with mitophagy was reported [19,20,21]. 

Mitophagy is a highly regulated multistep process in charge of the selective degradation of damaged/dysfunctional mitochondria by autophagy [22] and shares common aspects with other types of selective autophagy (including aggrephagy, ER-phagy, pexophagy, and xenophagy) [23,24,25]. This is vital for maintaining mitochondrial homeostasis and contributes, with other adaptive responses, to improving mitochondrial quality [20]. Therefore, it acts together with other pathways involved in the mitochondrial turnover, such as mitochondria-derived vesicles (MDVs), piecemeal mitophagy, and outer-mitochondrial, membrane-associated degradation (OMMAD) [20,26]. 

Basal, programmed, and stress-induced mitophagy, as well as transmitophagy are a few examples of how the term “mitophagy” can be interpreted differently according to the physiological and pathological context [22]. Basal mitophagy, the selective removal of damaged/aged mitochondria under steady-state conditions, was recently demonstrated by in vivo evidence [27,28,29]. In mouse tissues, it would operate as a housekeeping process, independently of exogenous triggers, to ensure mitochondrial homeostasis [30]. However, its levels change according to the specific tissue taken into consideration and between different cell types of the same tissue. Therefore, based on the energy demand, the basal levels of mitophagy are experienced at low (e.g., thymus) and high rates (e.g., heart and kidney) [27,31].

Programmed mitophagy plays a crucial role in physiological contexts which require the activation of the planned clearance of mitochondria as an element of their developmental programs, including tissue differentiation and protection processes [32,33]. In this regard, mitochondrial degradation was implicated in erythrocyte maturation through NIP3-like protein X (NIX/BNIP3L) activity and platelet activation [34,35]. In addition, the mitophagy-dependent selective clearance of paternal mitochondrial DNA (mtDNA) ensures the inheritance of the maternal mtDNA, following fertilization [36]. Programmed mitophagy occurs during cardiomyocytes and myoblasts maturation, macrophage polarization and retinal ganglion cell (RGC) differentiation, regulating the rewiring of metabolic pathways and the rate of mitochondrial bulk renewal [22].

Stress-induced mitophagy evolved to support the regulation of cell metabolism to external insults. It can be upregulated in response to different types of stresses, such as starvation, exercise, and ischemia [37,38,39]. It is essential in highly specialized cells primarily dependent on oxidative metabolism for aerobic respiration, such as skeletal muscle cells, neurons, adipocytes, cardiomyocytes, and hepatocytes [40,41]. Importantly, hypoxia was also described as driving mitophagy activation in these cells. The mitochondrial mass reduction would allow adaptive cellular responses to anaerobic conditions [42]. 

Although mitophagy was considered as a cell-autonomous function, transcellular mitochondrial degradation processes (transmitophagy) may also occur [43,44]. The presence of transmitophagy in the basal ganglia and its involvement in integrated neuron–astrocyte interplay are becoming increasingly clear [45]. 

Over the past decade, our knowledge of the molecular mechanisms regulating mitophagy has significantly advanced, and the involvement of multiple and redundant pathways is becoming increasingly evident. The best-characterized mechanism of mitophagy induction is represented by the PTEN-induced putative kinase 1 (PINK1)/Parkin-dependent pathway [46,47,48], given the relevance of these factors in autosomal recessive PD [49,50]. Although relevant in mitochondria removal in vitro as a result of specific damage, such as membrane depolarization, this pathway would seem unnecessary for governing basal mitophagy (in mice) in vivo, as well as other forms of selective autophagy [29,30]. In addition, in mice and *Drosophila melanogaster* deficient for either PINK1 or Parkin, the clearance of dysfunctional mitochondria still occurs [29,51,52]. Therefore, the Parkin-independent pathways (“mitophagy receptors” and “new ubiquitin E3 ligases”) were also reported to mediate the tethering of mitochondria to the autophagy apparatuses, directly activating mitophagy and operating to compensate for, or be in parallel to, PINK1–Parkin axis [22,40,53,54]. In this sense, signaling molecules (e.g., FUNDC1, BNIP3, BNIP3L/NIX, BCL2-L-13 and FKBP8) or lipids (e.g., cardiolipin) act as mitophagy receptors in responding to the environment and developmental stimuli.

On the other hand, impaired and excessive mitophagy are implicated in the pathogenesis of various ageing-associated diseases, including cancer, myocardial injury, diabetes, liver disease, sarcopenia, and neurological disorders such as Alzheimer’s and Parkinson’s diseases (AD and PD) [55,56]. Therefore, promoting mitophagy by pharmacological strategies may be a therapeutic target for diseases associated with mitochondrial dysfunction. 

The adenosine5′-monophosphate (AMP)-activated protein kinase (AMPK) (a highly conserved Ser/Thr kinase) is described as a master sensor of cell stress and is emerging as a crucial regulatory factor of mitochondrial metabolism and mitophagy [57]. Indeed, its implication in mitochondrial fission and autophagosomal engulfment, as well as its possible interplay with PINK1–Parkin signaling, was reported. In the present review, we examine recent advances in elucidating the molecular mechanisms that regulate mitochondrial removal and summarize our current knowledge on the signaling pathways governing mitophagy. Although the contributions of mitophagy in normal physiology and human disease are well known, these are beyond the scope of this review. In this regard, several reviews are available that illustrate the role of mitophagy in liver, cardiovascular, immune, inflammatory, and metabolic diseases and cancer [58,59,60,61,62]. Considering the direct connection between AMPK and mitochondrial biology, we also highlight recent advances in the role of the AMPK/ULK1 axis in regulating mitophagy, as well as its possible interplay with PINK1–Parkin signaling.

## 2. Regulation of Mitophagy

### 2.1. Chemical and Natural Mitophagy Triggers

In cultured cell lines, it is experimentally advantageous to use chemical reagents to trigger PINK1–Parkin-mediated mitophagy. In this regard, different compounds were described. An in vitro pharmacological manipulation allowed the dissection of molecular mechanisms underlying selective autophagy, where the acute loss in mitochondrial membrane potential (∆Ψ_m_) and the following stabilization of PINK1 could trigger the clearance of damaged mitochondria. Therefore, many protonophores (e.g., (carbonyl cyanide m-chlorophenyl hydrazone) CCCP/(carbonylcyanide-p-trifluoromethoxyphenylhydrazone) FCCP) and/or inhibitors of the respiratory chain (e.g., antimycin A/oligomicin) are widely used in biological research to promote mitochondrial depolarization and induce mitophagy via PINK1–Parkin pathway [63]. However, it is important to note that both CCCP and FCCP have some limitations as they exhibit cytotoxicity and are incorporated not only into the IMM (inner mitochondrial membrane), but also in the lysosome membrane, resulting in cytoplasmic acidification [63]. Therefore, new mitophagic inducers working independently of ∆Ψ_m_ dissipation and/or Parkin activation were identified. In addition, given the implication of mitophagy in MQC and in a wide variety of ageing-associated diseases, pharmacological approaches that aim to regulate mitophagy are currently receiving significant attention. In this regard, many synthetic chemical compounds (e.g., novel mitochondrial uncoupling agents, oxidative stress inducers, iron chelators, NAD^+^ precursors, and deubiquitinating enzymes and p53 inhibitors) and naturally occurring substances (e.g., spermidine, resveratrol A, urolithin A, and antibiotics) were shown to induce strong mitophagic responses [64,65].

### 2.2. PINK1–Parkin Axis

The best-known mitochondrial stress signaling pathway is the PINK1–Parkin-driven mitophagy, which mediates the specific ubiquitination and subsequent removal of dysfunctional/damaged mitochondria by recruiting autophagic machinery [46,66]. The Ser/Thr PINK1 (encoded by PARK6) and the cytosolic Parkin RING (Really Interesting New Gene)/HECT (Homologous to E6-AP Carboxyl Terminus) hybrid ligase (encoded by PARK2) are the critical factors of these molecular mechanisms. They are both mutated in autosomal-recessive PD, suggesting a causative role of impaired PINK1/Parkin-mediated mitophagy in the etiology of the disease [50,67]. Generally, the PINK1–Parkin pathway implicates the stabilization of PINK1 on the outer mitochondrial membrane (OMM), followed by the recruitment and activation of Parkin [47,68]. In turn, activated Parkin promotes the ubiquitination of a group of OMM proteins as well as the recruitment of autophagy adaptors, including OPTN (optineurin) and NDP52 (nuclear dot protein 52kDa) to damaged mitochondria [69,70,71,72], resulting in the constitution of LC3-associated phagophores, which leads mitochondria to lysosomal degradation. 

### 2.3. Structure and Activation of Parkin

Parkin belongs to the RBR (RING-between-RING) E3 ubiquitin family and adopts HECT mechanisms for substrate ubiquitination [73]. It contains a Ubl (N-terminal ubiquitin-like) domain followed by a flexible linker (a series of 60 amino acids), and four zinc-binding RING domains, namely RING0, RING1 (the E2-binding site), IBR (in-between-RING), and RING2 [74,75,76]. The REP (Repressor Element of Parkin), a short α-helix, is localized between the IBR and RING2 domains. All of these domains are conserved across the metazoans. Parkin acts like an RBR/HETC-type E3 ligase by promoting the ubiquitin transfer from an E2 ubiquitin-conjugating enzyme, UbcH7, UbcH8, and Ubc13/Uev1a, onto the catalytic center Cys431 in the RING2 domain, through the formation of an unstable thioester intermediate (transthiolation). Subsequently, ubiquitin is transferred from Parkin to an amino group of a substrate protein (acyl transfer step) via the lysine residue [77,78].

Under steady-state conditions, Parkin adopts an auto-inhibited conformation in which the RING0 domain partially masks Cys431, and both the Ubl domain and REP block the E2-binding site on RING1. Accordingly, the cytosolic ligase is greatly inactivated. 

It is now clear that PINK1-mediated phosphorylation on Ser65 residues of both ubiquitin (mono-Ub or multiple-attached Ubs in a chain at the mitochondrial surface), and the Ubl domain promotes the recruitment of latent Parkin onto the mitochondria, as well as its conversion from an autoinhibited enzyme to fully activated E3 ubiquitin ligase [79,80]. Therefore, PINK1 plays a role as ubiquitin kinase. A recent review goes into depth on the structural findings regulating Parkin activity [81]. PINK1 acts as a mitochondrial damage sensor, and when its import is arrested (under bioenergetics stress, loss in ∆Ψ_m_), it accumulates on the OMM of dysfunctional mitochondria and phosphorylates Ub molecules (pUb). The interaction between pUb and RING1 (at a site consisting of His302 and Arg305) of Parkin leads to the disengagement of the Ubl domain from the core structure, resulting in the conformational rearrangements, which in part liberate Parkin from the inhibitory interactions and promote its accumulation at the surface of mitochondria [76,82,83,84,85,86,87,88,89]. Therefore, the pUb acts as a receptor for Parkin recruitment [90], and the phosphoserine binding on RING1 govern Parkin localization. Subsequently, the Ubl domain of the partially activated configuration of Parkin is recognized and phosphorylated by PINK1 at Ser65. This promotes the interaction between pUbl and RING0 (in a pocket composed of the residues Lys161, Arg163, and Lys211) with the subsequent release of RING2 from RING0-mediated repression, and REP [79,80]. As a result, the opening of both E2-binding and catalytic sites enable Parkin to receive Ub from Ub-E2 with the subsequent ubiquitination of Lys residues in target OMM proteins. Parkin regulates the assembly of typical (at Lys48 and Lys63) and atypical ubiquitin chains (at Lys6 and Lys11) to the protein substrates involved in different aspects of mitochondrial functioning [85,91,92]. A large number of proteins undergo Parkin-mediated ubiquitination, including Mfn1/2, Miro1, subunits of the TIM/TOM complex and proteasomes, VDAC1/2/3 (voltage-dependent anion channel-1/2/3), metabolic proteins, HK1 (hexokinase-1), and factors of autophagy [93,94,95,96]. In total, it was indicated that more than 2000 proteins could be substrates of Parkin [94]. 

### 2.4. The Feed-Forward Mechanism of Parkin Activation

The addition of new Ub onto the OMM generates a virtuous circle since the presence of more substrates for PINK1 further increases the content of poly-phosphorylated Ub, leading to the extra activation and recruitment of Parkin to mitochondria. This results in a self-amplifying, feed-forward loop that, at the end, directs mitochondria along the mitophagy pathway. This model elucidates the low substrate selectivity of Parkin, which would be advantageous for advancing the positive feedback cycle, as opposed to its good spatial specificity for impaired mitochondria and the observation that Parkin activity is necessary for its recruitment [85,97,98,99,100]. Under basal conditions, it was suggested that other mitochondrial E3 ligases constitutively ubiquitylate mitochondrial substrates (e.g., MITOL/March5, localized at the TOM complex). Following mitochondrial depolarization, the MITOL-mediated ubiquitination would contribute to the early step in mitophagy, introducing the “initial Ub” required for Parkin recruitment and activation and the subsequent generation of the positive feedback cycle [101]. 

### 2.5. Molecular Links between PINK1/Parkin-Mediated Mitophagy and Mitochondrial Dynamics

The extensive catalytic activity of the PINK1–Parkin axis suggests the involvement of this pathway in several cellular processes. Therefore, the functional interplay between mitochondrial dynamics and trafficking, and PINK/Parkin system in ensuring the clearance of impaired mitochondria was reported. Indeed, Mfn2 and Miro (a mitochondrial motor/adaptor complex constituent) are targeted by PINK1 and Parkin [102,103]. In mammals, Mfn2 regulates mitochondrial fusion dynamics and ER–mitochondrial tethering [104]. When phosphorylated by PINK1 (at Ser378), Mfn2 adopts a fusion-nonpermissive conformation [105]. Moreover, the PINK1-mediated phosphorylation of Mfn2 on Ser442 and Thr111 enables Parkin–Mfn2 binding, presumably determining the initial recruitment of Parkin to mitochondria, the inhibition of Mfn2-regulated fusion and, ultimately, the activation of the mitophagic mitochondrial clearance [103,106].

The Parkin-mediated ubiquitination of Mfn1/2 leads to its proteasomal degradation by a p97-dependent extraction mechanism [107,108,109,110]. Thus, Mfn2 degradation leads to the interdiction of fusion and the induction of fission events as well as to the separation of mitochondria–ER contact sites [110]. The Mfn2 localization in the mitochondria–ER contacts and the detection of PINK1 in MAM (mitochondria-associated membrane) also suggest a PINK1/pUb-mediated Parkin recruitment at the mitochondria–ER sites [111]. Different from the autophagy pathway, mitophagy may exhibit an antagonistic and reciprocal relationship with mitochondria–ER contacts as their reduction is functional to the Parkin-mediated ubiquitination of substrates, its recruitment to the OMM and mitochondrial turnover [111]. 

Other studies indicate that Miro is phosphorylated by PINK1 at multiple amino acid residues, including Ser156. This promotes Miro ubiquitination by Parkin and its subsequent proteasomal degradation [102,112,113,114]. In this way, Miro degradation and Mfn2 inhibition block mitochondrial motility and fusion, respectively, thus favoring mitochondrial clearance by autophagosome. 

### 2.6. Deubiquitinating Enzymes and PTEN-L as Regulators of Mitophagy

The ubiquitination process is reversible and balanced by the activities of deubiquitinating enzymes (DUBs), which modulate protein turnover by removing Ub from ubiquitinated substrates. Many DUBs, such as USP15 (Ub-specific cysteine protease 15), USP30, USP33 and USP35 regulate mitochondrial homeostasis and antagonize PINK1-Parkin-driven mitophagy [92,115,116,117,118]. In this regard, a mechanism of action of USP30-mediated K6-linkage-specific deubiquitination was suggested [119,120]. USP30 is localized in the OMM, and it is physically associated with TOM complex components [121]. It also regulates the TOM–mitochondrial import by a sort of “tug of war” with MITOL. Therefore, USP30-induced low levels of ubiquitination, charged to the specific site of PINK1 stabilization, and may dampen the initial triggering responsible for the PINK/Parkin-mediated amplification process of ubiquitination [122,123]. On the contrary, USP8 acts as a positive regulator of Parkin-driven mitophagy, since the specific removal of K6-linked Ub conjugates from Parkin stimulates its turnover and speeds up the mitophagy process [124]. Although USP30 is traditionally believed to be a negative regulator of Parkin-mediated ubiquitination, its ablation in USP30 knockout cells did not affect the ubiquitination of Parkin substrates [121,125]. The picture of the complexity of the deubiquitination reactions is enriched by further elements, since the PINK1-mediated phosphorylation of Ub alters Ub structure and compromises the enzymatic activities of USP8, USP15, and USP30 [126]. USP35 detaches from depolarized mitochondria and impairs mitophagy with a mechanism independent of Parkin recruitment [127]. On the contrary, upon mitochondrial depolarization USP15 is recruited on the OMM and removes K48- and K63-polyUb chains from protein substrates [116]. A recent study revealed USP33, located at the OMM, as a novel Parkin deubiquitinase; it is able to antagonize Parkin pro-mitophagy activity [118]. USP33 modulates Parkin self-ubiquitination at Lys435 and specifically removes K6-, K11-, K63- and K48-linked Ub chains from Parkin.

Interestingly, a negative regulation of mitophagy by PTEN-long (PTEN-L), a novel protein phosphatase that acts dephosphorylating pSer-Ub, was recently shown. This activity keeps Parkin in the auto-inhibited conformation and inhibits its recruitment and activation in the OMM. In addition, PTEN-L has the ability to target pSer-Ub chains, leading to the impairing of the feed-forward mechanism of mitophagy [128,129].

### 2.7. pUbl-Independent Mechanism of Parkin Activation, the Unexpected Plasticity of RING0 Binding Site

One unsolved question regards the existence of molecular mechanisms other than PINK1-mediated phosphorylation for Parkin activation. The positive feed-forward amplification cycle of mitophagy also regulates Parkin in a phosphorylation-independent manner. Several studies demonstrated that Parkin recruitment to damaged mitochondria was not completely eliminated through the absence of the Ubl domain or Ser65 [85,130,131,132]. Moreover, the in vitro activation of unphosphorylated Parkin by pUb was observed, but the underlying mechanism still remains unclear [83].

A new study shows the pUb’s ability to directly activate Parkin in the absence of the phosphorylation of its Ubl domain (Figure 1) [133]. This secondary mechanism for the activation of Parkin would be dependent on the RING0 pUbl-binding site. The binding of pUb to the RING0 pUbl-binding site enables pUb to act as a signal for Parkin recruitment to OMM and activation (Figure 1A). In this regard, RING0 would have a higher affinity for pUb than the pUbl domain. Similar to the PINK1-mediated activation of Parkin, the association of the second pUb to RING0 (Figure 1B-ii), instead of the pUbl domain (Figure 1B-i), leads to conformational reorganization, in which the catalytic RING2 domain is released. This mechanism reveals an amazing flexibility of RING0 for binding pUb and pUbl, suggesting the possibility that Parkin localization and activation may be controlled by other phosphoproteins that have the ability to bind RING0 or RING1 domains [133]. Thus, other activation mechanisms were proposed, but they require the support of structural analysis in future works. It was reported that Parkin increased its activity following Polo-like, kinase-mediated phosphorylation at Ser378. In turn, Parkin activation leads to the ubiquitination of important regulators of mitosis, Cdh1 and Cdc20 [134]. Furthermore, Parkin phosphorylation at Ser131 and Tyr143 was catalyzed, respectively, by CDK5 (cyclin-dependent kinase), and c-Abl (Abelson tyrosine-protein kinase) reduced its ubiquitination activity [135,136,137].

### 2.8. PINK1 Processing and Stabilization

In functional mitochondria with an unimpaired ∆Ψ_m_, newly synthesized PINK1 is imported across the IMM by the outer- and inner-membrane translocase complexes, TOM and TIM, via the positively charged mitochondrial-targeting sequence. Therefore, PINK1 interacts with the surface receptors TOM20, TOM22, and TOM70; subsequently, it passes through the TOM40 translocation pore and IMM TIM23 complex [98,138,139]. Following the import, PINK1 is clipped in multiple ways: first, by the IMM-resident α-cleaved form of PINK1/PGAM5-associated, rhomboid-like protease (PARL) within its transmembrane domain (between Ala103 and Phe104 residues); and second, by the mitochondrial processing peptidase (MPP) in the matrix [140,141,142,143]. The *m*-AAA protease AFG3L2, embedded in the IMM and with the active site facing the matrix, may act cooperatively with PARL, supporting its action [144]. The resulting cleaved (52kDa fragment) PINK1 has the Phe104 in its N-terminus that acts as a targeting signal (type-2 N-degron) for the cytosolic E3 ubiquitin ligases (UBR1, UBR2, and UBR4). Thus, PINK1s (the short form of PINK1) is released into the cytoplasm, marked by ubiquitin, and ultimately degraded via the N-end rule pathway [145]. In this context, it was also proposed that the PINKs anchored in the OMM is polyubiquitinated and that Lys137 residue, instead of Phe104, plays a critical role in promoting PINK1 ubiquitination and its successive degradation via the proteasome system [146]. Nonetheless, the activation of the PINK1 kinase is prevented. 

On the other hand, PINK1 is a sensor of mitochondrial damage, and when activated, triggers the mitophagy signal. PINK1 import is arrested upon mitochondrial uncoupling, and the enzyme is no longer subjected to protease cleavage. This results in the stabilization and activation of full-length (63 kDa) PINK1 in the OMM [47,48]. In this sense, the critical regulators for PINK1 import arrest and accumulation on the OMM are the OMS (OMM localization signal) domain of PINK1 and TOM7 [139], a subunit of TOM40 complex. To confirm this, PINK1 is imported and cleaved by OMA1 in depolarized mitochondria when TOM7 is absent [139]. Although the specific role of TOM7 is still unknown, it may be implicated in mediating the lateral release of the kinase from the TOM40 channel. Beyond the OMS domain, a specific negatively charged motif in PINK1 is also required to regulate the import [139]. As PINK1 builds on the OMM, its kinase activity is increased by dimerization and subsequent auto-phosphorylation at Ser228 and Ser402, which presumably occur in the TOM40 complex [147,148,149]. These steps enable PINK1 to phosphorylate the Ub and Ubl domain in Parkin. In addition to this, other mitochondrial proteins were recently reported to be phosphorylated by PINK1, including Mic60/Mitofilin and NDUFA10 [150,151].

## 3. Functions of Mitophagy Receptors 

Under physiological and pathological conditions, mitophagy can occur independently of the presence of Parkin. These pathways rely on the intervention of receptors (BNIP3, NIX, FUNDC1, BCL2L13, FKBP8) constitutively localized in the OMM (via the C-terminal transmembrane (TM) domain) and containing a conserved LIR (LC3-interacting region) motif (at N-terminal region), which allows their association with phagophore on its LC3-decorated membrane.

(a)BNIP3 and NIX

Although BNIP3 and NIX were initially reported to promote apoptosis and necrosis [152,153,154,155], these proteins are now well known to induce mitophagy. For instance, in response to hypoxia, they are similarly upregulated by HIF1α to mediate mitophagy [34,156,157,158]. Stress signals induce both BNIP3 homodimerization (via its TM domain) and its integration into the OMM, essential processes for the efficient removal of mitochondria [157]. BNIP3-LC3 interactions require BNIP3 phosphorylation at Ser17 and Ser24 [159]. In addition, myocyte lipotoxicity can activate NIX-mediated mitophagy. In this context, NIX-induced mitophagy is reversed by the PRKA/PKA-catalyzed phosphorylation of NIX at Ser212 (within the TM domain), leading to the retrotranslocation of the receptor from the sarcoplasmic reticulum and mitochondria to the cytosol [160]. NIX is involved in ensuring mitochondrial clearance during reticulocyte development [161] and is also required for mediating mitophagy following ischemia-reperfusion in the brain [158]. In CCCP-exposed cells, NIX mediates mitophagy by promoting GABARAP-L1 recruitment on the OMM and the binding of the latter with its LIR motif [162]. ROS (reactive oxygen species) generation induces the translocation of Rheb (a small GTPase) to mitochondria resulting in NIX/LC3 complex formation, which promotes the formation of mitophagosomes. It is noteworthy that the phosphorylation at the Ser34 and Ser35 sites near the LIR domain increases NIX affinity to LC3A/B and induces mitophagy [163]. NIX dimerization, which is modulated by C-terminal NIX phosphorylation, paired with BNIP3, is critical for initiating mitophagy [164]. A new study demonstrates that the kinase responsible for the phosphorylation of Ser17 in BNIP3 and Ser35 in NIX is ULK1 while TBK1 may be implicated in phosphorylating Ser24 and Ser34 in BNIP3 and NIX, respectively [165]. NIX and BNIP3 also act on multiple levels in PINK/Parkin-mediated mitophagy. In particular, BNIP3 contributes to PINK1 stabilization, facilitates the translocation of DRP1, and prevents BECN1/BCL-2 complex formation [156,166,167]. Parkin-mediated ubiquitination of NIX promotes NBR1 interaction with Ub and LC3/GABARAP leading to the generation of autophagosomes surrounding damaged mitochondria [168]. In depolarized mitochondria, NIX contributes to Parkin accumulation [169].

(b)FUNDC1

Hypoxia-induced mitophagy is also regulated by the activity of the mitophagy receptor FUNDC1, which contains an LIR motif and three TM domains [170]. Under hypoxia conditions, mitochondrial E3 ligase MITOL can regulate the protein expression levels of FUNDC1 [171]. In order to avoid uncontrolled mitophagy, MITOL-mediated ubiquitination of FUNDC1 at Lys119 targets the receptor for proteasomal degradation [171]. FUNDC1 phosphorylation on residues Tyr18 and Ser13 near the LIR domain also contributes to regulating FUNDC1-LC3 interactions. Under basal conditions, FUNDC1 is inactivated through the phosphorylation of Tyr18 and Ser13 residues by Src tyrosine kinase and CK2 (casein kinase2), respectively. Hypoxia conditions inactivate Src causing the reduced phosphorylation of Tyr18, FUNDC1 binding with LC3, and the formation of the mitophagosome [170]. Under stress conditions, the dephosphorylation of Ser13 by Ser/Thr phosphatase PGAM5 stabilizes FUNDC1-LC3 interactions, promoting mitophagy [172]. By contrast, PGAM5-FUNDC1 interactions are inhibited during normoxia conditions, as BCL2L1/Bcl-xL is associated with phosphatase, preventing the dephosphorylation of FUNDC1 Ser13 and mitochondria removal [173]. Interestingly, an enhanced association between FUNDC1 and LC3 is induced by the ULK1-mediated phosphorylation of FUNDC1 at Ser17 near the LIR domain [174]. During cardiac progenitor cell differentiation, FUNDC1- and NIX-mediated mitophagy is essential in mediating the remodeling of the mitochondrial network [175]. 

(c)BCL2L13, FKBP8, AMBRA-1

In mammalian cells, additional autophagy receptors were studied, although were less well characterized, including BCL2L13, FKBP8 and AMBRA-1. BCL2L13 seems to mediate mitophagy by regulating mitochondrial fission [176] and by interacting with ULK1 [177]. FKBP8 mediates mitochondrial fragmentation and Parkin-independent mitophagy by the recruitment of LC3A [178]. AMBRA1 (autophagy and Beclin 1 regulator 1) can associate with LC3 via an LIR motif and can also induce Parkin-independent and -dependent mitophagy [179,180]. In SH-SY5Y cells, AMBRA1-mediated mitophagy protects against apoptosis induced by redox status alterations [181]. 

### 3.1. Other Promoters of Mitophagy: Cardiolipin and Novel E3 Ligases 

Recent findings identified cardiolipin (a dimeric phospholipid of the IMM) as a positive regulator of mitophagy [182]. It can translocate to the OMM upon mitochondrial damage and associate with LC3 to promote mitophagy [182,183]. 

In addition, novel regulators of Parkin-independent mitophagy were recently identified, including other E3 ligases such as MUL1, ARIH1, CIAP1/2, Gp78, and TRAF2, and the Synuclein alpha-interacting protein (Synphilin-1) [184,185,186,187,188,189]. In cancer cells, where ARIH1 (Ariadne RBR E3 Ub protein ligase 1) is highly expressed and Parkin is often downregulated, the Ub ligase activity of ARIH1 promotes mitophagy in a PINK1-dependent manner, contributing to chemotherapy resistance [188]. In cultured cells and rat brain tissues, the PINK1-mediated recruitment of synphilin-1 to mitochondria promotes the SIAH1-mediated ubiquitination of mitochondrial substrates, which results in the recruitment of LC3 for autophagosome formation [187]. Finally, in a PD model MUL1 can rescue the PINK1- and Parkin-mutant phenotypes, compensating for PINK1/Parkin loss [186].

### 3.2. Autophagy Adaptors Linking to PINK1–Parkin-Mediated Mitophagy

During Ub-mediated selective autophagy, including PINK1–Parkin-mediated mitophagy, autophagy receptors such as p62/SQSTM1, NBR1, NDP52, Tax1BP1, and OPTN play crucial roles in the recognition and autophagic encapsulation of target cargoes [190]. The autophagy adaptors have both an LIR domain and Ub-binding region that interact with the Ub chains associated with the cargoes. They are all phosphorylated by TBK1 (TANK-Binding Kinase1), increasing their binding affinity to Ub-chains and ATG8 proteins [191,192]. In particular, OPTN is phosphorylated on Ser177 to enhance the binding affinity with LC3 [193] and on Ser 473 and Ser 513 to enhance its ubiquitin-chain binding activity [191]. 

### 3.3. The Linear Sequence of Mitophagosome Synthesis

Canonical mitophagy implicates the selective engulfment of damaged mitochondria into autophagosomes. In turn, autophagosomes quickly fuse with lysosomes allowing for complete mitochondrial degradation. Historically, mitophagophore formation was thought of as a linear sequence described below.

Mitophagosome initiation requires the coordinated recruitment of the ULK complex (composed of ULK1, FIP200, ATG13, and ATG101) and ATG9-containing vesicles on ER-associated membranes [194,195]. These regions are focal points for the subsequent recruitment of the vacuolar protein sorting (VPS)34 complex (consisting of the phosphatidylinositol 3-kinase (PI3K) VPS34, and the adaptors ATG14, VPS15, and Beclin1), which produces PI3P [196,197] on DFCP1 (double FYVE-containing protein 1)-positive ER segments known as “omegasomes” [198]. Omegasomes amplify the local PI3P signals, thus attracting WIPI1/2 that recruit additional ATGs (including ATG5-12-16), as well as the lipidation machinery that covalently modifies ATG8s (LC3 and GABARAP subfamilies) with PE (phosphatidylethanolamine) [199,200]. As part of the autophagosomal membrane, the PE-conjugated ATG8 family acts as a tether between cargo and phagophore [201]. Mitophagy then requires the elongation and maturation of the phagophore through lipids addition. Once the endosomal sorting complex is in place, the autophagosome is closed and sorted towards the lysosomes [202]. 

However, recent studies revealed some vulnerabilities in this model that are discussed later in more detail.

### 3.4. Mitophagosome Synthesis: Redundancy and Positive Feedback Signals, and the Role of mTORC1

Mitophagosome synthesis is far from being a linear sequence of events since it seems to involve amplification loops and feedback signals [191]. In this sense, recent studies also revealed a different sequence of reactions with respect to starvation-induced autophagy (mTORC1-ULK1-LC3), a different implication of mTORC1, and new ways to fine-tune mitophagy processes. In addition, reports suggest that mitophagy depends not only on pre-existing membranes but also on the de novo synthesis of the autophagosome via the activation of early components. 

Although mTORC1 is the master regulator during starvation-induced autophagy [203,204], its involvement during mitophagy has not yet been clarified. Some studies suggested that PINK1–Parkin may require the inactivation of mTOR [205,206]. Consistent with this, the inhibition of PINK1–Parkin-mediated mitophagy is induced by mTORC1 hyperactivation [206,207,208]. Additionally, hypoxia conditions can reduce mTORC1 activity [209]. On the other hand, during Invermectin-induced mitophagy, mTORC1 activity does not seem to change [55,189,210], and mTORC1 overexpression does not affect selective autophagy [71]. The functional significance of LC3 during mitophagy is also still debatable, as well as the coordinated recruitment of early autophagy structures (ULK complex and ATG9 unit) on ER-associated membranes. For instance, in cardiomyocytes and HeLa cells, a non-canonical mitophagy pathway not requiring LC3 lipidation was reported to occur upon starvation and hypoxia [38,211]. Following the loss in ∆Ψ_m_, Uthe LK1 complex and ATG9 unit are independently associated with dysfunctional mitochondria, even when membrane-bound LC3 is lacking [194], and in an adaptor-dependent manner [190].

Although all five autophagy receptors can mediate the recruitment of ATG8-associated membranes during mitophagy, only OPTN and NDP52 are crucial for mitochondrial clearance. The generation of phospho-Ub signals on the OMM is fully dependent on PINK1 but does not fully require Parkin activity. Therefore, PINK1 is able to recruit NDP52 and OPTN independently from Parkin. In this sense, PINK1 acts as both a damage sensor and effector, whereas Parkin would *amplify* rather than trigger the mitophagy signal [190]. NDP52 recruitment to ubiquitinated mitochondria is independent of LC3, but it is enough for the initiation of phagophore [212]. Indeed, OPTN and NDP52 can significantly *amplify* the mitophagy via the ATG8-dependent *positive feedback loop* [213]. During mitophagy, the existence of multiple initiation events was suggested by the fact that ATG13 and OPTN could interact with mitochondria in a discontinuous oscillatory modality [189]. NDP52 promotes both the recruitment and activation of ULK1 on cargo through the association of the NDP52-FIP200/ULK1 complex [71]. Additionally, in addition to binding ATG8 proteins, OPTN interacts with ATG9A molecules. Therefore, the recruitment of both OPTN and NDP52 on the ubiquitinated mitochondria are critical in promoting de novo autophagosomal membrane formation, almost to Ub-positive cargo. Therefore, Parkin-mediated mitophagy seems to rely on the NDP52-ULK1/LC3 and OPTN-ATG9/LC3 sequence of reactions. 

### 3.5. Autophagosome Elongation and Closure

Mitophagy requires the expansion of phagophore membranes. The recruitment of GTPase-activating proteins (GAPs) for initiating RAB7 cycling plays a critical role in this process. Cargo ubiquitination by Parkin recruits RABGFE1, which triggers the endosomal Rab GTPase cascade, including RAB5 and MON1/CCZ1 [214]. In turn, MON1/CCZ1 facilitates the recruitment of RAB7A to mitochondria, and RAB7A promotes ATG9 vesicle assembly to the autophagosome formation area. TBC1D15/D17 and TBC1D5 act as mitochondrial GAPs for Rab GTPase and target ATG8/ATG9 family proteins around damaged mitochondria by modulating Rab7 activity, in order to recruit the isolation membrane [215,216]. TBK1-mediated phosphorylation of RAB7A assists in bringing ATG9 vesicles to mitochondria [217]. Thus, these findings suggest the central role played by the endosomal Rab cycles during mitophagy. The removal of dysfunctional mitochondria requires autophagosome–lysosome fusion. Although the loss of ATG8 family members does not abrogate autophagosome formation, ATG8 proteins are essential for autophagosome–lysosome fusion [212] and degradation of the inner autophagosomal membrane [218].

### 3.6. AMPK/ULK1 Axis in Mitophagy Cascade

In response to mitochondrial damage, as well as under energetic stress, the AMPK complex (consisting of a catalytic α, a scaffolding β, and a regulatory γ subunit) acts as a sensor. It engages downstream effectors implicated in metabolic processes, autophagy, and in different aspects of mitochondrial homeostasis, including biogenesis, dynamics, and, ultimately, the clearance of damaged mitochondria, to restore homeostasis. Canonically, a radical increase in the cellular AMP/ATP ratio triggers the full activation of AMPK due to AMP/ADP binding to the γ subunit and the subsequent phosphorylation of the Thr172 by the upstream kinase liver kinase B1 (LKB1) [57]. The direct link of AMPK-mediated, energy-sensing function to the autophagy process is represented by the Ser/Thr kinase ULK1 [219]. In addition to ULK1, AMPK also interacts with ATG9 and components of the Class III PI3K complex 1 of the autophagy pathway [57]. ULK1 represents the most upstream activator of the autophagic pathway and is extensively phosphorylated by AMPK [203,220,221,222]. ULK1 is a core constituent of the autophagy pre-initiation complex with ATG101, ATG13, and FIP200. All of these components are substrates of ULK1 kinase activity as well as ULK1 itself. Additional targets of ULK1 were reported, including AMBRA1, VPS34, syntenin-1, TAB2, Raptor, FUNDC1, BNIP3, ATG14, ATG16L, Sec16a, and Sec23a [165,174,223,224,225,226,227,228,229,230,231,232]. In addition to the autophagy activation, it is becoming more evident that the AMPK/ULK1 axis plays a crucial role in promoting mitophagy [233,234]. Indeed, ULK1- or AMPK-deficient cell lines exhibit an increased accumulation of morphologically altered mitochondria, suggesting that the AMPK-dependent phosphorylation of ULK1 is crucial for the selective clearance of mitochondria [220,235,236]. In this regard, AMPK-mediated ULK1 phosphorylation at Ser555 regulates ULK1 translocation to mitochondria and mitophagy [234,237]. As the proton ionophore CCCP is a strong inducer of PINK1–Parkin-mediated mitophagy and AMPK by ATP depletion, new studies are developing in order to establish which relationship links PINK/Parkin pathway with the AMPK/ULK1 axis. In this regard, exactly how Parkin first senses problems with mitochondria and how specific phosphorylation events orient towards mitophagy remain to be defined [238]. Another important question is how AMPK regulates mitophagy in skeletal muscle. In this regard, new PINK1–Parkin-independent mechanisms and the activation of specific, subcellular AMPK pools were reported [239,240,241]. 

### 3.7. The Early Role of the AMPK/ULK1 Axis in Triggering the Rapid Activation of Parkin 

As described above, although it is well known that Parkin can sense mitochondrial stress and promote mitophagy, the initial input in dictating the earliest Parkin recruitment remains to be clarified. In this sense, a model revealing the implication of the AMPK-ULK1 axis in starting the first translocation of Parkin onto the OMM was recently proposed in vivo and in vitro (Figure 2A) [238]. In particular, Parkin was reported as a novel ULK1 substrate. Under mitochondrial stress (exposure to CCCP or valinomycin), the immediate activation of cytosolic AMPK (within 2 min) leads to the phosphorylation of its downstream substrates, including ULK Ser555, Raptor Ser 792, MFF Ser146, and ACC Ser79. At the same time, ULK1 activation results in the specific phosphorylation of Parkin Ser108 (P-Parkin-108), an event that seems to be localized in the cytoplasmic region [238]. This phosphorylation falls in a new highly conserved region named the ACT element (activating element). This short region is localized in the flexible linker between the UBL and RING0 domains and is suggested to be critical for Parkin activation (Figure 2A, Top) [79]. Interestingly, phosphoproteomic analyses previously revealed the phosphorylation of Parkin Ser108 in brown fat, although the kinase responsible for this event was not identified identified [242]. In parallel to P-Parkin108, phosphorylations of Beclin Ser30 and ATG16L1 Ser278, two ULK1 substrates [224,230], were also observed. Within 10 min of CCCP exposure, the recruitment of AMPK and ULK1 to mitochondria leads to MFF phosphorylation and mitochondrial fission. Only at later time points (after 30 min) does the phosphorylation of Parkin Ser65 occur. This event coincides with the ubiquitination of the substrates of Parkin, CISD1 and Mfn2, and TBK activation at Ser172 [238]. This study demonstrates that the rapid and greatest PINK1-mediated phosphorylation of Parkin Ser65 requires the ULK1-dependent phosphorylation of Parkin at Ser108. In addition, these findings highlight the crucial and early role played by the AMPK/ULK1 axis in mitophagy and place Parkin regulation downstream of AMPK/ULK1, revealing a new route to modulating Parkin. The direct association of Parkin with ULK1, and thus AMPK, enriches the scenario of possible connections between AMPK and mitochondria. 

### 3.8. Other Scenarios of AMPK- and ULK1-Mediated Mitophagy

The interplay between PINK1–Parkin signaling and AMPK was also demonstrated in cardiomyocytes [243]. Therefore, the AMPKα2-mediated phosphorylation of PINK1 at Ser495 stimulates the PINK1–Parkin–SQSTM1 signaling implicated in cardiac mitophagy, and these events are essential for preventing the progression of heart failure. Parkin was also reported as an AMPK substrate [244]. In particular, Parkin is phosphorylated (at Ser9) and activated by AMPK to trigger RIPK3 polyubiquitination and reduce cell death during necroptosis.

The direct assembly of the autophagosome to the deranged mitochondrion is another mechanism by which AMPK is able to regulate mitophagy, independently of ULK1 [233]. In particular, the association of the AMPK complex with dysfunctional mitochondria, via the N-myristoylation of AMPKß subunits, promotes the recruitment of the ATG16 complex to the damaged regions and triggers the assembly of the autophagosome. Additionally, in leukemia stem cells (LSCs), the mitochondrial stress derived from oncogenic transformation triggers AMPK/FIS1-mediated mitophagy in an effort to keep the healthy mitochondria needed for LSCs self-renewal and survival [245]. In hepatocytes, AMPK upregulates mitophagy by increasing UQCRC2 (ubiquinol-cytochrome c reductase core protein 2) protein expression levels through the activation of NFE2L2/NRF2 (nuclear factor, erythroid like 2) [246]. 

ULK1, for its part, is an active element of the mitophagic integrated system and its crucial role was determined. In some cases, it may additionally connect mitophagy modulators with the PINK/Parkin system. Indeed, ULK1 can directly phosphorylate mitophagy receptors such as FUNDC1, BNIP3, NIX, BCL2L13 and VCP/p97 [165,174,177]. In particular, VCP/p97 is recruited to regions affected by Parkin ubiquitination. The ULK1-mediated phosphorylation of FUNDC1 at Ser17 promotes FUNDC1/LC3 interaction and mitophagy. In mammalian cells, BCL2L13 induces mitophagy by forming a complex with ULK1 and LC3B [176,177]. Moreover, ULK1 specifically promotes mitophagy by phosphorylating NIX and BNIP3 on Ser 35 and Ser17, respectively, to stimulate their interaction with LC3. The phosphorylation of BNIP3 also leads to its stabilization due to the inhibition of proteasomal turnover [165].

### 3.9. The Mitochondrial Pool of AMPK (mitoAMPK) Governs the Spatial Specificity of Energetic Stress-Induced Mitophagy 

AMPK-mediated ULK1 phosphorylation at Ser555 regulates mitophagy in different cell types, including human acute myeloid leukemia stem cells, and mouse skeletal muscle [237,247] In particular, it was demonstrated that a single bout of exercise triggers mitophagy in skeletal muscle during the recovery period [237] in a PINK1-independent manner [37]. The direct phosphorylation of the mitochondrial fission factor (MFF), which triggers mitochondrial fragmentation before mitophagy, by AMPK also demonstrates a critical role of the kinase in mitochondrial dynamics [248]. Zong et al. [249] demonstrated that distinct thresholds of energetic stress can trigger different compartmentalized pools of AMPK. In addition, different heterotrimeric complexes of AMPK may show specificity for different targets suggesting that this spatial activation is the key for managing energy stress. Consistent with this, Drake et al. [241] demonstrated the presence of specific isoforms of mitoAMPK in different tissues in mice and humans. In particular, the isoforms of AMPK α1, α2, β2, and γ1 are physically associated with the OMM, and this mitochondrial localization is conserved in human skeletal muscle and heart. Additionally, different energetic stresses (e.g., ischemia, muscle contractions, and treadmill running) can activate mitoAMPK in vivo and this event is required for mitophagy (Figure 2B). Indeed, the inhibition of mitoAMPK activity reduces exercise-induced mitophagy.

In skeletal muscle, AMPK is emerging as a crucial regulator of mitophagy, while experimental evidence linking the PINK1–Parkin pathway to the selective removal of mitochondria is limited [240]. Thus, in C2C12 myotubes, energetic stress induces AMPK-mediated mitochondrial fission via the direct phosphorylation of MFF, and TBK1 phosphorylation, possibly via ULK1 activated by AMPK, in a PINK1–Parkin-independent way. In this context, AMPK promotes mitophagy by increasing, on the one hand, mitochondrial fission, and on the other, TBK1-mediated mitochondrial autophagosome engulfment (Figure 2C [239]).

## 4. Conclusions and Future Perspectives

Over the past decade, our understanding of the molecular mechanisms governing mitophagy has considerably increased, not only due to the levels of complexity of the pathways involved but also because a lack of mitophagy was associated with numerous aging-related diseases. Although detailed mechanistic studies revealed the importance of PINK1–Parkin signaling in regulating mitophagy, multiple molecular players and redundant pathways are also involved in managing mitochondrial removal. Thus, it is now clear that the modulation of basal mitophagy does not require the involvement of the PINK1–Parkin axis, which may be more related to the mitochondrial removal in response to specific stresses. In renewing our understanding of the molecular mechanism underlying mitophagy, the emerging role of AMPK is also becoming increasingly crucial in light of its ability to promote the early and major step of mitophagy (by the sequence AMPK-ULK1-Parkin). PINK1 and Parkin are activated by complete mitochondrial depolarization while AMPK and ULK1 require mild mitochondrial perturbations (e.g., metformin). Therefore, it will be interesting to delineate the role and function of the AMPK/ULK1 axis in the activation of Parkin. Indeed, they are shared elements in different diseases including neurodegeneration, cancer and diabetes. This may allow the expansion of the homeostatic and/or pathogenic conditions in which Parkin plays a critical role in maintaining mitochondrial health. Altogether, these issues may aid to design novel therapeutic approaches to target pathophysiological processes correlated with mitochondrial dysfunction. 

## Figures and Tables

**Figure 1 cells-11-00030-f001:**
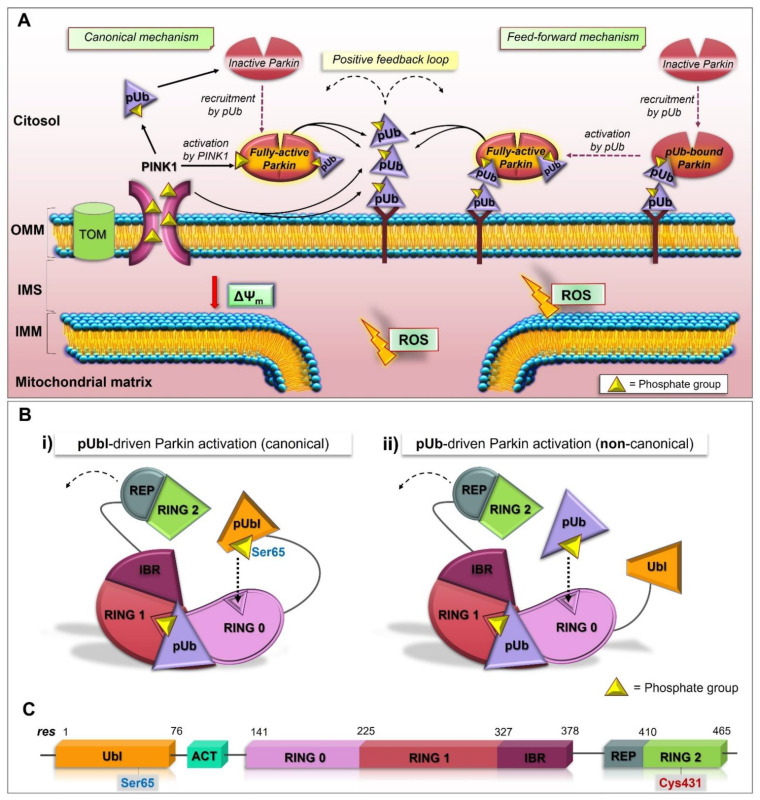
A working model representing simultaneous canonical and feed-forward mechanisms of Parkin activation. (**A**) Canonical mechanism: PINK1 activates Parkin directly by phosphorylating Parkin Ubl domain at Ser65 and indirectly through phosphorylation of Ub proteins at Ser65, relieving the autoinhibition of Parkin ligase. This enables Parkin to poly-ubiquitinate substrates on the OMM. These chains are then phosphorylated by PINK1 and act as docking sites for further recruitment and activation of Parkin (positive feedback loop); feed forward mechanism: PINK1-mediated phosphorylation of Ub on damaged mitochondria leads to the recruitment of latent Parkin. A second pUb acts as a fixed activation signal of Parkin leading to the conformational rearrangements which release its autoinhibition. Parkin-mediated addition of new Ub on the OMM gives rise to positive feedback cycle. OMM: outer mitochondrial membrane; IMS: intermembrane space; IMM: inner mitochondrial membrane. (**B**) Mechanisms of Parkin activation. (i) pUbl- or (ii) pUb-binding to RING0 domain causes conformational rearrangements resulting in REP and RING2 release determining fully activated Parkin. (**C**) Schematic representation of primary structure of Parkin with Ser65 phosphorylation site and Cys431 catalytic residue in Ubl and RING2 domains, respectively.

**Figure 2 cells-11-00030-f002:**
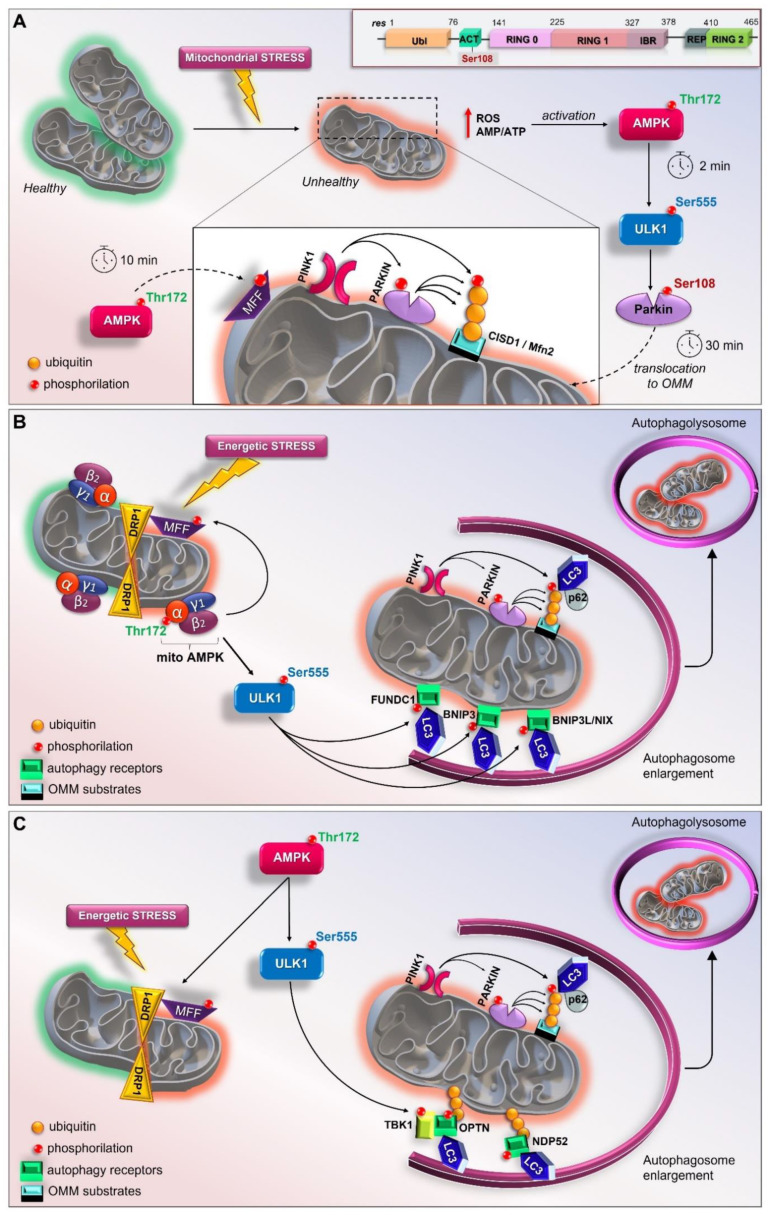
New current models of AMPK/ULK1-mediated mitophagy. (**A**) Top, schematic representation of primary structure of Parkin showing ACT element with Ser108 residue. In mouse livers and primary hepatocytes AMP/ULK1 axis triggers the rapid activation of Parkin (by Hung et al.) [238]. Upon mitochondrial depolarization, AMP/ATP imbalance and increases in mtROS immediately activate AMPK/ULK1 signaling. Therefore, in the cytoplasm ULK1 phosphorylates Parkin at Ser108 within 2 min of depolarization treatment. Later, AMPK phosphorylates MFF to induce mitochondrial fission and PINK1 phosphorylates Parkin at Ser65; (**B**) In skeletal muscle, mitoAMPK regulates the spatial specificity of mitophagy in a context of mitochondrial remodeling (by Drake et al.) [241]. Following energetic stress, mitoAMPK is activated in vivo and promotes ULK1-mediated formation of autophagosome which fuse with lysosomes to allow the complete mitochondrial degradation; (**C**) In C2C12 myotubes, AMPK coordinates mitophagy through mitochondrial fission via MFF and TBK1-mediated autophagosomal engulfment via ULK1 activation (by Seabright et al.) [239].

## Data Availability

Not applicable.

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
