# Peer review of "Mitophagy: Molecular Mechanisms, New Concepts on Parkin Activation and the Emerging Role of AMPK/ULK1 Axis"

_cells, 2021, doi:10.3390/cells11010030_

Round 1
Reviewer 1 Report
The review by the Dr. Roberto et al. discusses one of the Parkin-independent pathways of mitophagy regulation: AMPK/ULK1. Here, they review the recent literature in molecular mechanisms and signalling pathways, focusing on the early role and spatial specificity of the AMPK/ULK1 axis. Overall, this review is very thorough, very nicely written and the quality of the visuals is excellent. I just want to draw the attention to inconsistencies with the shrift, changing from 10 to 11.5 , first leaving the impression that it is on purpose r.20 and r.103 and then switching completely randomly, e.g. r. 143-147;154-163;166-174;192-248;259-263;etc.
Author Response
We thank the reviewers for their useful comments, that strengthen our work. Please find below a point-to-point answer to their questions. We have modified the title, the abstract and the text to respond to all the issues and elaborated on the changes below. We have also improved the review with the Graphical Abstract.
All coauthors have agreed to the revisions.
REVIEWER#1
The review by the Dr. Roberto et al. discusses one of the Parkin-independent pathways of mitophagy regulation: AMPK/ULK1. Here, they review the recent literature in molecular mechanisms and signalling pathways, focusing on the early role and spatial specificity of the AMPK/ULK1 axis. Overall, this review is very thorough, very nicely written and the quality of the visuals is excellent. I just want to draw the attention to inconsistencies with the shrift, changing from 10 to 11.5 , first leaving the impression that it is on purpose r.20 and r.103 and then switching completely randomly, e.g. r. 143-147;154-163;166-174;192-248;259-263;etc.
Response: This problem does not depend on us but is due to the file conversion by the system during the submission processing
Reviewer 2 Report
In this manuscript Roberto et al provide a comprehansive overview of the current knowleadge on the molecular mechanisms of mitophagy.
The manuscript is well written and provide sufficient details and updates.
I have only minor comments:
- The main part oft he review is related tot he PINK1-Parkin axis rather than AMPK-ULK1 axis. Therefore, the title may be modified.
- Lines 43-49: unneccessasry text
- Line 46: better „ROS generation“ instead of „ROS dissemination“
- Introduction: The Introduction requires a clear structure and should be improve. The author may apply the following structure:
- what is mitophagy
- basal mitophagy
- stress-induced mitophagy
- regulation of mitophagy
- translational significance the mitophagy (i.e., relevance for diseases)
- Paragraph 2.1 fits better just before paragraph 2.13
- Combine the paragraph 2.3 with 2.4
- Line 212: „…when its import is impaired…“ One should explaine under which conditions its import is impired
- Line 262: Please explain the significance of the mitochondria-ER contact for mitophagy
- Reduced fusion is not nesseccery stimulates mitophagy, please correct
- Please explain the not commonly used abbreviations
Author Response
We thank the reviewers for their useful comments, that strengthen our work. Please find below a point-to-point answer to their questions. We have modified the title, the abstract and the text to respond to all the issues and elaborated on the changes below. We have also improved the review with the Graphical Abstract.
All coauthors have agreed to the revisions.
REVIEWER#2
In this manuscript Roberto et al provide a comprehansive overview of the current knowleadge on the molecular mechanisms of mitophagy.
The manuscript is well written and provide sufficient details and updates.
I have only minor comments:
- The main part of the review is related to the PINK1-Parkin axis rather than AMPK-ULK1 axis. Therefore, the title may be modified.
Response: We partially agree with the reviewer’s comment. In the review we have reported recent advances in elucidating the canonical molecular mechanisms and signaling pathways regulating mitophagy by also focusing on mechanisms of Parkin activation and the early role and spatial specificity played by AMPK/ULK1 axis. We believe on the great importance of molecular mechanisms regarding Parkin recruitment and activation independently from PINK1 functions and the novel findings about the role of AMPK in mitophagy. On this basis we have modified the title as follow: “Mitophagy: molecular mechanisms, new concepts on Parkin activation and the emerging role of AMPK/ULK1 axis”.
- Lines 43-49: unneccessasry text
Response: Lines 43-49 have been eliminated
- Line 46: better „ROS generation“ instead of „ROS dissemination“
Response: Lines 43-49 have been eliminated
- Introduction: The Introduction requires a clear structure and should be improve. The author may apply the following structure:
- what is mitophagy
- basal mitophagy
- stress-induced mitophagy
- regulation of mitophagy
- translational significance the mitophagy (i.e., relevance for diseases)
Response: We have improved the structure of the introduction following the reviewer’s suggestion
- Paragraph 2.1 fits better just before paragraph 2.13
Response: The order of the paragraphs has been modified
- Combine the paragraph 2.3 with 2.4
Response: Paragraph 2.3 has been eliminated
- Line 212: „…when its import is impaired…“ One should explaine under which conditions its import is impired
Response: we have modified the sentence as follow: …PINK1 acts as a mitochondrial damage sensor and when its import is arrested (under bioenergetics stress, loss in ∆Ψm)……
See the revised section - Structure and activation of Parkin
- Line 262: Please explain the significance of the mitochondria-ER contact for mitophagy Reduced fusion is not nesseccery stimulates mitophagy, please correct
Response: See the revised section - Molecular links between PINK1-Parkin-mediated mitophagy and mitochondrial dynamics
…….”Differently from the autophagy pathway, mitophagy may exhibit an antagonistic and reciprocal relationship with mitochondria-ER contacts as their reduction is functional to Parkin-mediated ubiquitination of substrates, its recruitment to the OMM and mitochondrial turnover [111]”….
- Please explain the not commonly used abbreviations
Response: Done
Reviewer 3 Report
The review “Mitophagy: new concepts on molecular mechanisms and the emerging role of AMPK/ULK1 axis” by Roberto et al. provides a good review on mitophagy with a focus on Pink/Parkin independent mechanisms of mitophagy. There are some comments and suggestions for improving the review below. However, my main point would be that there is already a multitude of recent reviews on mitophagy out there. This one tries to focus on AMPK/ULK1 and PINK/Parkin independent mechanisms, but I am not entirely convinced there is a completely different angle here that has not yet been covered elsewhere.
Main points:
- The Text shifts between different font types – please correct.
- While generally, it is well written and has a good structure, there are sections that require copy-editing and grammar corrections.
- Not sure paragraph 2.3 is needed.
- Sections 2.11, 2.13 and 3.2 are not very clear and would benefit from rephrasing. For instance, section 2.11 is very long and I wonder if this could be restructured with a common underlying theme.
- According to the title, section 3 is the main part of the review. However, it is relatively short compared to the overall text. I think overall the review does not focus on AMPK/ULK1, but rather on Pink/Parkin independent mitophagy. Maybe that could be better reflected in the title.
Author Response
We thank the reviewers for their useful comments, that strengthen our work. Please find below a point-to-point answer to their questions. We have modified the title, the abstract and the text to respond to all the issues and elaborated on the changes below. We have also improved the review with the Graphical Abstract.
All coauthors have agreed to the revisions.
REVIEWER#3
- The Text shifts between different font types – please correct.
Response: This problem does not depend on us but is due to the file conversion by the system during the submission processing
- While generally, it is well written and has a good structure, there are sections that require copy-editing and grammar corrections.
Response: Done
- Not sure paragraph 2.3 is needed.
Response: Paragraph 2.3 has been eliminated
- Sections 2.11, 2.13 and 3.2 are not very clear and would benefit from rephrasing. For instance, section 2.11 is very long and I wonder if this could be restructured with a common underlying theme.
Response: we have restructured and improved sections 2.11, 2.13 and 3.2.
See the revised sections:
- Functions of mitophagy receptors
- Mitophagosome synthesis: redundancy and positive feedback signals, and the role of mTORC1
- Other scenarios of AMPK- and ULK1-mediated mitophagy
- According to the title, section 3 is the main part of the review. However, it is relatively short compared to the overall text. I think overall the review does not focus on AMPK/ULK1, but rather on Pink/Parkin independent mitophagy. Maybe that could be better reflected in the title.
Response: We partially agree with the reviewer’s comment. In the review we have reported recent advances in elucidating the canonical molecular mechanisms and signaling pathways regulating mitophagy by also focusing on mechanisms of Parkin activation and the early role and spatial specificity played by AMPK/ULK1 axis. We believe on the great importance of molecular mechanisms regarding Parkin recruitment and activation independently from PINK1 functions and the novel findings about the role of AMPK in mitophagy. On this basis we have modified the title as follow: “Mitophagy: molecular mechanisms, new concepts on Parkin activation and the emerging role of AMPK/ULK1 axis”.
Round 2
Reviewer 3 Report
This review has improved and I am overall satisfied with the corrections. There are still some minor things that need changing:
Line 133: replace "ambit" with "scope"
Line 217: replace "are" with "could be"
The fonts are still inconsistent (e.g. line 178, line 372, line 474).
In Figure 2C, the bottom panel does not mention which site of AMPK is phosphorylated (but the top panel does). Please add.
Author Response
We thank the reviewer for his/her useful comments, that strengthen our work. Please find below a point-to-point answer to the questions. We have modified the text and the figure 2C to respond to all the issues and elaborated on the changes below.
All coauthors have agreed to the revisions.
This review has improved and I am overall satisfied with the corrections. There are still some minor things that need changing:
Line 133: replace "ambit" with "scope"
Response: Done
Line 217: replace "are" with "could be"
Response: Done
The fonts are still inconsistent (e.g. line 178, line 372, line 474).
Response: Unfortunately this problem does not depend on us. It is only due to the file conversion by the system during the submission processing.
In Figure 2C, the bottom panel does not mention which site of AMPK is phosphorylated (but the top panel does). Please add.
Response: Done